# PROMPT-AWARE ADAPTER: TOWARDS LEARNING EFFECTIVE VISUAL TOKENS FOR GPT4-STYLE MULTIMODAL MODELS

## ABSTRACT

The rapid advancement of Large Language Models (LLMs) has revolutionized chatbot systems, resulting in unprecedented levels of intelligence. Moreover, the recent GPT4-style models have demonstrated extraordinary multi-modal abilities, such as generating human-like responses based on visual inputs and textual prompts. To bridge the gap between the vision and language modalities, GPT4-style models usually learn an adapter that converts the visual inputs to understandable tokens for LLMs. However, those adapters are usually independent of textual prompts, thus outputting invariant visual tokens, regardless of the question of interest. Those prompt-irrelevant visual tokens significantly increase the burden of visual reasoning on LLMs. In this paper, we propose prompt-aware adapter, which is equipped with an ability of dynamically embedding visual inputs based on the prompt. In this way, the proposed adapter extracts the most informative visual clues to the prompt, thus largely facilitating LLMs for visual understanding. Experiments on various questions, including object classification, color recognition, counting and position reasoning, demonstrates the effectiveness of the proposed method. Code will be publicly available.

## 1 INTRODUCTION

Large Language Models (LLMs) (Brown et al., 2020; Chung et al., 2022; OpenAI, 2023; Touvron et al., 2023; Chiang et al., 2023) have recently demonstrated remarkable success in natural language processing. With unprecedented language understanding and logic reasoning capabilities, these models can perform a variety of intricate linguistic tasks, such as text summarization, question answering, dialogue processing and writing new essays or articles. Since then, there has been a growing interest among academics and researchers to develop various LLMs for different fields, and simultaneously explore the extensions into Multimodal Large Language Models (MLLMs) (Li et al., 2023b; Dai et al., 2023; Liu et al., 2023; Li et al., 2023c; Zhang et al., 2023).

To equip LLMs with the visual perception ability, most MLLMs employ a trainable adapter that aligns a frozen visual encoder with a frozen LLM. For example, MiniGPT-4 (Zhu et al., 2023) and LLaVA (Liu et al., 2023) employs a simple linear layer to convert visual features into readable tokens for LLMs. Those adapters show an effective ability to translate visual signals, especially for simple scenes and straightforward questions. However, when trying to perceive complex scenarios or parse complicated questions or prompts, directly projecting visual features may be not sufficient because LLMs need to carefully watch the entire image and then select the most informative hints by themselves from visual tokens for reasoning.

In other words, when parsing the visual context, LLMs have to pay the equal attention to every detail of an image. However, because visual encoders and adapters are designed to emit a fix number of visual features and tokens, it is difficult to carry every detail with only a limited number of tokens. In this case, LLMs may fail to appropriately or correctly capture the necessary information for answering. Moreover, regions of interest in an image to different questions may vary significantly. As shown in Fig. 1 (a), to answer "what is sitting on the chair?" the visual tokens should be mainly related to the "chair" and "dog". For the question "what color is the umbrella?" the visual tokens

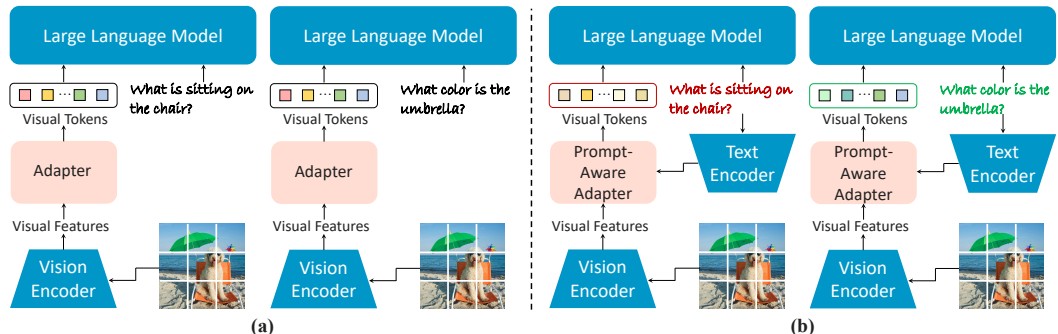

Figure 1: Comparison between prompt-unaware and prompt-aware adapters. **(a)** Prompt-unaware adapters. Translated visual tokens are independent of prompts (questions of interest) and thus are the same. **(b)** Prompt-aware adapters adaptively embeds the informative clues for visual reasoning based on the prompt. Therefore, visual tokens are different.

should focus on the "umbrella?" However, in the two cases, most existing adapters emit the same visual tokens, which significantly increase the burden of LLMs for visual parsing.

In this paper, we propose prompt-aware adapter, which is a family of adapters that are able to adaptively pick up the informative clues for visual reasoning based on the prompt. Our motivation is that, when converting visual features into readable tokens, prompt is used to guide adapters. In this way, adapters are aware of prompts, thus able to know how to uncover visual context and shift the attention to the regions of interest. Compared to prompt-unaware adapters, our adapters not only project visual features but also filter information, which significantly facilities visual perception for LLMs. As shown in Fig. 1 (b), given the sampe context image, prompt-aware adapters are able to adjust the visual tokens based on prompts.

There can be many ways to implement a prompt-aware adapter. In this paper, our adapter is constructed as follows. First, similar to Q-Former (Li et al., 2023b), we create a set number of learnable query embeddings as input to the adapter. Then, a visual transformer that interacts with the frozen visual encoder for visual feature extraction through cross-attention layers. Third, a textual transformer is inserted into the adapter and interacts with the visual features based on prompts. In this way, our adapter is able to sense prompts and capture the informative details for answering questions. To evaluate the proposed prompt-aware adapter, we conduct experiments on various questions, including object classification, color recognition, counting and position reasoning. The results show the effectiveness of the proposed method. Specifically, our method improves the accuracy of above four perception tasks by 2.89%, 5.80%, 1.43% and 2.98%, respectively. In summary, our contributions are as follows.

- Our research reveals that prompt-independent adapters may be not sufficient to capture the most informative visual clues for visual understanding.
- Based on Transformers with cross-attention, we propose a prompt-aware adapter, which is equipped with an ability of dynamically embedding visual inputs based on prompts.
- Experiments on various questions show that the proposed adapter effectively improves the visual reasoning ability for MLLMs.

## 2 RELATED WORK

**Large Language Models.** It has been found that scaling up model size or data size of Pre-trained Language Models (PLMs) can often improve model capabilities on downstream tasks. The development of PLMs based on Transformer (Vaswani et al., 2017) architecture, such as BERT (Devlin et al., 2018), GPT-2 (Radford et al., 2019), T5 (Raffel et al., 2020), etc., has laid a solid foundation for the emergence of LLMs. As one of the most well-known LLMs, GPT-3 (Brown et al., 2020) increases the model parameters to 175B, achieving impressive performance on zero-shot or few-shot downstream tasks. It has inspired the development of various LLMs, including InstructGPT (Ouyang

et al., 2022), LaMDA (Thoppilan et al., 2022), OPT (Zhang et al., 2022), PaLM (Chowdhery et al., 2022), Flan-T5 (Chung et al., 2022) and so on. A remarkable application of LLMs is ChatGPT (OpenAI, 2023), which adapts the GPT series of LLMs for dialogue, showcasing its amazing conversation capability. In addition, benefiting from the strong base of LLaMA (Touvron et al., 2023), a large number of high-quality open-source LLMs have been spawned, such as Alpaca (Taori et al., 2023), Vicuna (Chiang et al., 2023), Llama 2 (Touvron et al., 2023), etc. Those LLMs show impressive emergence on natural language processing tasks, but they cannot process visual information.

**Multimodal Large Language Models.** Large vision backbone models have made rapid progress in perceiving visual information (Kirillov et al., 2023; Caron et al., 2021; Oquab et al., 2023; Radford et al., 2021), but have developed slowly in terms of inference. Combining the perception ability of vision models and the reasoning ability of LLMs brings the new field of Multimodal Large Language Models (MLLMs).The MLLMs are primarily image-to-text generative models. The general components of current MLLM includes a vision encoder, a connection module and a LLM decoder (Li, 2023). To bridge the gap between vision and language, Flamingo (Alayrac et al., 2022) adds Perceiver Resampler and inserts gated cross-attention dense blocks. With its ability for vision-language contextual learning, Flamingo is often considered as the GPT-3 moment in the multimodal domain. BLIP-2 (Li et al., 2023b) and InstructBLIP (Dai et al., 2023) achieve modal proximity by training only a lightweight Q-Former as the connection module. LLaVA (Liu et al., 2023) connects image features to word embedding space through a simple linear layer. MiniGPT-4 (Zhu et al., 2023) utilizes both frozen pre-trained Q-Former and trainable linear layer to make visual tokens understandable by LLMs along with textual tokens. VisionLLM (Wang et al., 2023) treats images as a foreign language, which are converted into tokens via a Language-Guided Image Tokenizer. It employs language instructions to flexibly define all tasks both in the vision and language domains. LLaMA Adapter Zhang et al. (2023) adds image tokens to a set of learnable adaption prompts. The adaption prompts are added to word tokens at higher transformer layer of LLaMA. Current connection modules (or adapters) produce the same visual tokens for different prompts, resulting in the general performance of MLLMs on logical reasoning tasks. Hence, we focus on designing the prompt-aware adapters, which can learn effective visual tokens tailored to specific prompts.

## 3 Proposed Method

In this section, we introduce the architecture of the proposed method in details. As shown in Fig. 2, the architecture consists of a vision encoder, a text encoder, a prompt-aware adapter and an LLM. First, the vision and text encoders extract the image features and prompt embeddings, respectively. Second, guided by prompt embeddings, the prompt-aware adapter projects image features into visual tokens, so that the LLM is able to understand the input images. Third, the LLM accepts the visual tokens and prompt embeddings and then emits the response.

**Vision Encoder.** Following existing MLLMs, we use a pre-trained vision encoder to extract image features. Specifically, given an image $I \in \mathbb{R}^{HW \times 3}$, where $H$ and $W$ are the height and width, the vision encoder produces the corresponding features $X \in \mathbb{R}^{M \times C_x}$. Usually, $M << HW$ and $C_x$ is the number of feature channels. In implementation, we use the ViT-g/14 from EVA-CLIP (Fang et al., 2023) as the vision encoder, where $M = 257$ and $C_x = 1408$. Note that, during training, the vision encoder is frozen, whose parameters are not updated.

**Text Encoder.** We use a pre-trained text encoder to process the input prompt. Specifically, given a prompt $T \in \mathbb{R}^{S \times 1}$, where $S$ is the length of the sentence, the text encoder embeds each word in the prompt and outputs the corresponding text embedding $Y \in \mathbb{R}^{S \times C_y}$. In this paper, we implement two variants of text encoder, including the BERT (Devlin et al., 2018) ($C_y = 768$) and the LLaMA (Touvron et al., 2023) encoder ($C_y = 5120$). We take BERT as an example when describing dimensions later. Similar to vision encoder, the text encoder is frozen during training.

### 3.1 Architecture

**Prompt-Aware Adapter.** As indicated in Fig. 2, the proposed prompt-aware adapter includes $N$ attention blocks and a single linear layer. Each attention block is composed of a self-attention layer, a visual cross-attention layer, a text cross-attention layer and a feed forward layer. The specific processing flow of the prompt-aware adapter is as follows.

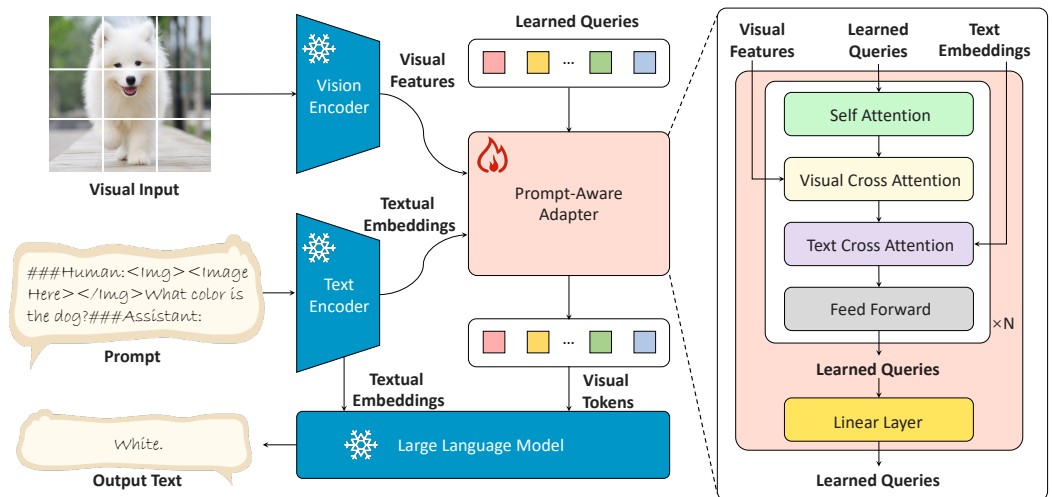

Figure 2: Architecture of the proposed method. First, the pre-trained vision encoder and text encoder are used to extract visual features and text embeddings, respectively. Then, the prompt-aware adapter produces variant visual tokens based on the prompt. Finally, the LLM accepts visual and promt tokens and prompt embeddings, and emit the response.

First, we create a few query tokens $\boldsymbol{Z} \in \mathbb{R}^{L \times C_z}$ through the standard normal distribution, where $L$ is the number of query tokens and $C_z$ is the dimension of query embeddings. The query tokens capture the dependencies between each other through the self-attention layer:

$$\boldsymbol{Q}_z = \boldsymbol{Z} \cdot \boldsymbol{W}_z^q, \quad \boldsymbol{K}_z = \boldsymbol{Z} \cdot \boldsymbol{W}_z^k, \quad \boldsymbol{V}_z = \boldsymbol{Z} \cdot \boldsymbol{W}_z^v,$$

$$\text{Self-Attention}(\boldsymbol{Q}_z, \boldsymbol{K}_z) = \text{softmax}\left(\frac{\boldsymbol{Q}_z \cdot \boldsymbol{K}_z^T}{\sqrt{C_z^k}}\right), \tag{1}$$

$$\boldsymbol{O}_z = \text{Self-Attention}(\boldsymbol{Q}_z, \boldsymbol{K}_z) \cdot \boldsymbol{V}_z,$$

where $\boldsymbol{W}_z^q \in \mathbb{R}^{C_z \times C_z^k}, \boldsymbol{W}_z^k \in \mathbb{R}^{C_z \times C_z^k}, \boldsymbol{W}_z^v \in \mathbb{R}^{C_z \times C_z^v}, \boldsymbol{Q}_z \in \mathbb{R}^{L \times C_z^k}, \boldsymbol{K}_z \in \mathbb{R}^{L \times C_z^k}, \boldsymbol{V}_z \in \mathbb{R}^{L \times C_z^v}$ and $\boldsymbol{O}_z \in \mathbb{R}^{L \times C_z^v}$. The $C_z^k$ and $C_z^v$ are the dimensions of key and value, respectively. In implementation, we set the number of query tokens $L$ to be 32. The $C_z$, $C_z^k$ and $C_z^v$ are set to 768.

Second, the learned queries $\boldsymbol{O}_z \in \mathbb{R}^{L \times C_z^v}$ retrieve relevant information from visual features $\boldsymbol{X} \in \mathbb{R}^{M \times C_x}$ with cross-attention:

$$\boldsymbol{Q}_x = \boldsymbol{O}_z \cdot \boldsymbol{W}_x^q, \quad \boldsymbol{K}_x = \boldsymbol{X} \cdot \boldsymbol{W}_x^k, \quad \boldsymbol{V}_x = \boldsymbol{X} \cdot \boldsymbol{W}_x^v,$$

$$\text{Visual-Cross-Attention}(\boldsymbol{Q}_x, \boldsymbol{K}_x) = \text{softmax}\left(\frac{\boldsymbol{Q}_x \cdot \boldsymbol{K}_x^T}{\sqrt{C_x^k}}\right), \tag{2}$$

$$\boldsymbol{O}_x = \text{Visual-Cross-Attention}(\boldsymbol{Q}_x, \boldsymbol{K}_x) \cdot \boldsymbol{V}_x,$$

where $\boldsymbol{W}_x^q \in \mathbb{R}^{C_z^v \times C_x^q}, \boldsymbol{W}_x^k \in \mathbb{R}^{C_x \times C_x^k}, \boldsymbol{W}_x^v \in \mathbb{R}^{C_x \times C_x^v}, \boldsymbol{K}_x \in \mathbb{R}^{M \times C_x^k}$ and $\boldsymbol{V}_x \in \mathbb{R}^{M \times C_x^v}$. The $\boldsymbol{O}_x \in \mathbb{R}^{L \times C_x^v}$ is the final output of the visual cross-attention layer. In implementation, we set $L = 32$, $M = 257$, $C_z^v = 768$, $C_x = 1408$, $C_x^k = 768$, and $C_x^v = 768$.

Third, queries $\boldsymbol{O}_x \in \mathbb{R}^{L \times C_x^v}$ interact with the prompt embeddings $\boldsymbol{Y} \in \mathbb{R}^{S \times C_y}$ via the text cross-attention.

$$\boldsymbol{Q}_y = \boldsymbol{O}_x \cdot \boldsymbol{W}_y^q, \quad \boldsymbol{K}_y = \boldsymbol{Y} \cdot \boldsymbol{W}_y^k, \quad \boldsymbol{V}_y = \boldsymbol{Y} \cdot \boldsymbol{W}_y^v,$$

$$\text{Text-Cross-Attention}(\boldsymbol{Q}_y, \boldsymbol{K}_y) = \text{softmax}\left(\frac{\boldsymbol{Q}_y \cdot \boldsymbol{K}_y^T}{\sqrt{C_y^k}}\right), \tag{3}$$

$$\boldsymbol{O}_y = \text{Text-Cross-Attention}(\boldsymbol{Q}_y, \boldsymbol{K}_y) \cdot \boldsymbol{V}_y,$$

where $\boldsymbol{W}_y^q \in \mathbb{R}^{C_x^v \times C_y^k}$, $\boldsymbol{W}_y^k \in \mathbb{R}^{C_y \times C_y^k}$, $\boldsymbol{W}_y^v \in \mathbb{R}^{C_y \times C_y^v}$, $\boldsymbol{Q}_y \in \mathbb{R}^{L \times C_y^k}$, $\boldsymbol{K}_y \in \mathbb{R}^{S \times C_y^k}$ and $\boldsymbol{V}_y \in \mathbb{R}^{S \times C_y^v}$. The $C_y^k$ and $C_y^v$ represent the key and value dimensions of the text features, respectively. We set the dimensions as follows: $L = 32$, $C_x^v = 768$, $C_y = 768$, $C_y^k = 768$, and $C_y^v = 768$. The output $\boldsymbol{O}_y \in \mathbb{R}^{L \times C_y^v}$ can be considered as soft visual prompts, conditioning the visual representation onto LLM.

Those tokens $\boldsymbol{O}_y \in \mathbb{R}^{L \times C_y^v}$ are then processed through the feed forward layer to obtain visual output $\boldsymbol{X}' \in \mathbb{R}^{L \times C_x'}$, where the $C_x' = 768$. Finally, to further complete the modal alignment, a single linear projection layer is appended at the end of the module, which maps the dimension $C_x''$ of visual feature $\boldsymbol{X}'' \in \mathbb{R}^{L \times C_x''}$ from 768 to 5120. Now, the dimension of $\boldsymbol{X}'' \in \mathbb{R}^{L \times C_x''}(32 \times 5120)$ is considerably smaller than the dimension of $\boldsymbol{X} \in \mathbb{R}^{M \times C_x}(257 \times 1408)$ obtained by the frozen vision encoder. In a certain sense, the prompt-aware adapter can be considered to filter out invalid information and only focus on the visual content of the specific question.

**Large Language Model Decoder.** After the prompt-aware adapter aligning image features to the text space, the LLM-based model harvests the ability to receive and reason with multi-modal information. To be specific, the pre-trained auto-regressive LLM is responsible for generating text sequences based on the given visual signals $\boldsymbol{X}'' \in \mathbb{R}^{L \times C_x''}$ and prompt embeddings $\boldsymbol{Y} \in \mathbb{R}^{S \times C_y}$. Note that $\boldsymbol{Y} \in \mathbb{R}^{S \times C_y}$ also need to be projected into $\boldsymbol{Y}' \in \mathbb{R}^{S \times C_y'}$ for dimension alignment with $\boldsymbol{X}'' \in \mathbb{R}^{L \times C_x''}$, i.e., $C_y' = C_x'' = 5120$. The training goal of LLM decoder is to maximize the likelihood probability $\sum_{\langle \boldsymbol{A}, \boldsymbol{X}'', \boldsymbol{Y}' \rangle \in \mathbb{D}} \log P(\boldsymbol{A}|\boldsymbol{X}'', \boldsymbol{Y}')$ on the training dataset $\mathbb{D}$. Finally, the probability distribution of the answer $\boldsymbol{A} \in \mathbb{R}^{K \times C_a}$ is output, where $K$ represents the number of generated tokens and the dimension $C_a$ represents the vocabulary size. In the concrete implementation, similar to MiniGPT-4 (Zhu et al., 2023), we use Vicuna (Chiang et al., 2023), which is built on LLaMA-13B (Touvron et al., 2023), as the LLM.

### 3.2 IMPLEMENTATION DETAILS

**Model Specification.** In the specific implementation of prompt-aware adapter, the number of attention blocks is set to 12. The visual cross-attention layer is inserted every other attention block, that is, at blocks 0, 2, 4, 6, 8, and 10. The text cross-attention layer is attached to blocks 7 to 10. means we introduce text signals to focus on relevant image features at later stages of visual queries.

**Datasets and Training.** First, our model loads the pre-trained parameters of MiniGPT-4 (Zhu et al., 2023) as initial weights. Specifically, the first pre-training stage of MiniGPT-4 covers more than 5 million image-text pairs including Conceptual Caption (Changpinyo et al., 2021; Sharma et al., 2018), SBU (Ordonez et al., 2011), and LAION (Schuhmann et al., 2021). The second fine-tuning stage of MiniGPT-4 utilizes 5,000 high-quality image-text pairs. Then, we unfreeze all layers of the original Q-Former as well as the linear layer for end-to-end fine-tuning on the COCO-QA (Ren et al., 2015) dataset. To be specific, COCO-QA is a large-scale question-answering dataset constructed on image captions of MS-COCO (Lin et al., 2014). It consists of a relatively balanced mix of logical questions and answers in areas including object classification, color recognition, counting and position reasoning.

In addition, the prompts used in the model are defined as the following format:

> *###Human: <ImgHere></Img><Prompt> ###Assistant:*

During training, the keyword *<ImgHere>* is replaced with the feature representation $\boldsymbol{X}'' \in \mathbb{R}^{L \times C_x''}$ of visual input, while ** and *</Img>* are retained as positioning markers and will not be replaced. In image captioning tasks, the identifier *<Prompt>* is randomly sampled from pre-defined prompts of various forms. For instance, we adopt text descriptions such as "Please provide a detailed description of the picture.", "Could you describe the contents of this image for me?", etc. as prompt. In visual question answering tasks, it is important to note that the keyword *<Prompt>* should be replaced with the specific question. For example, regarding the COCO-QA dataset, the *<Prompt>* is substituted with questions related to visual input, such as "What color is the umbrella?" or "What is sitting on the chair?" and so on. More specific information about constructing prompts is shown in Appendix A.1.

**Hyper-parameter Settings.** The model applies the AdamW (Loshchilov & Hutter, 2017) optimizer with $\beta_1 = 0.9$, $\beta_2 = 0.999$, and a weight decay rate of 0.05. We linearly warm up the learning rate from $10^{-6}$ to $3 \times 10^{-5}$ in the first 200 steps to make the model converge quickly. Then, we perform cosine decay with a minimum learning rate of $10^{-5}$. The maximum number of training epochs is set to 200, with 200 iterations per epoch. The batch size during training is 16. As a result, all models are trained using a single NVIDIA A100 (80G) GPU and are completed within 6 hours.

## 4 EXPERIMENTS AND ANALYSIS

In this section, we evaluate the perception and cognition capabilities of the proposed model as well as provide relevant analyses. Among them, perception ability includes recognizing the existence, color, quantity and location of objects. Cognition ability is typically manifested in tasks such as common-sense reasoning. First, Section 4.1 quantitatively evaluates the perception capability of the proposed model. Then, we provide the qualitative evaluation of perception, cognitive and generalization ability in Section 4.2. Finally, Section 4.3 analyzes the current limitations and possible directions for improvement.

### 4.1 QUANTITATIVE EVALUATION

**Perception Tasks and Datasets.**

The open-ended answers from MLLMs presents considerable challenges to the quantization (Fu et al., 2023). Existing methods tend to use GPT (OpenAI, 2023) or manual scoring (Li et al., 2023a; Liu et al., 2023), which may suffer from inaccuracy and subjectivity. Hence, when conducting quantitative evaluations, we only adopt questions with accurate and concise answers. Specifically, the COCO-QA dataset (Ren et al., 2015) that is based on image-text pairs of MS-COCO (Lin et al., 2014) are used in the quantitative assessment. Specifically, the dataset covers 123,287 images, 112,684 training question-answer pairs, and 5,000 test question-answer pairs. Table 1 shows the data distribution used for end-to-end fine-tuning and testing in four types of perception tasks. The training data accounts for around 95.75%, and the remaining question-answer pairs are used for testing on the zero-shot image-to-text generation task. Furthermore, the proportion of question categories in the training and test data is almost identical.

Table 1: Distribution of question types for model perception ability assessment.

| Task Category | Train | Percentage | Test | Percentage |
|---|---|---|---|---|
| Object Classification | 78,666 | 69.81% | 3,532 | 70.64% |
| Counting | 1,8761 | 16.65% | 807 | 16.14% |
| Color Recognition | 8,304 | 7.37% | 336 | 6.72% |
| Position Reasoning | 6,953 | 6.17% | 325 | 6.50% |
| Total | 112,684 | 95.75% | 5,000 | 4.25% |

**Zero-shot Evaluation.** We apply the prompt-tuned model to zero-shot image-to-text generation. The generated responses are subsequently compared to the ground truth to calculate metrics. We focus on evaluating the performance of model on four perception tasks, including object classification, quantity counting, color recognition and position reasoning. To be specific, we separately evaluate the accuracy of generated answers on each task, as well as the overall accuracy.

The comparison is performed between the proposed model and the following three models: (1) Original two-stage pre-trained MiniGPT-4 model. (2) MiniGPT-4 only fine-tunes the linear layer on the COCO-QA dataset, while keeping the Q-Former frozen. (3) MiniGPT-4 simultaneously unfreezes the Q-Former and the linear layer for fine-tuning on the COCO-QA dataset.

Table 2 illustrates the quantitative results on test dataset. MiniGPT-4 has achieved satisfactory results even before fine-tuning on the object classification task, indicating its solid understanding of object types. Furthermore, the results indicate that current GPT-style models are not sensitive to the quantity of objects, even after fine-tuning in downstream tasks. The proposed prompt-aware adapter is capable of selectively focusing on objects mentioned in the question during visual feature extraction, leading to a noticeable improvement (approximately 50%) in quantity perception. In the color recognition and positional reasoning tasks, improving the adapter structure leads to accuracy improvements of around 1.43% and 8.18%, respectively.

Table 2: Quantitative results of object classification, counting, color recognition and position reasoning on the test dataset. Total Acc (%) refers to the overall accuracy on the four tasks.

| Method | Object Classification Acc (%) | Counting Acc (%) | Color Recognition Acc (%) | Position Reasoning Acc (%) | Total Acc (%) |
|---|---|---|---|---|---|
| MiniGPT-4 | 73.69(±0.69) | 36.01(±2.64) | 63.32(±0.77) | 57.23(±1.08) | 68.42(±0.28) |
| Fine-tuned (Frozen) | 76.38 (±0.90) | 67.85 (±0.42) | 75.34 (±0.64) | 60.92 (±2.69) | 74.96 (±0.26) |
| Fine-tuned (Open) | 75.25 (±0.77) | 64.88 (±1.80) | 77.19 (±0.48) | 62.15 (±1.22) | 74.20 (±0.51) |
| **Ours** | **78.65** (±0.14) | **72.03** (±0.95) | **78.31** (±0.71) | **67.69** (±1.70) | **77.26** (±0.28) |

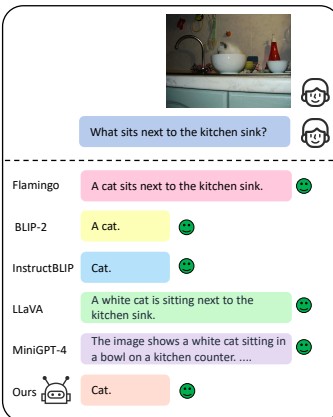
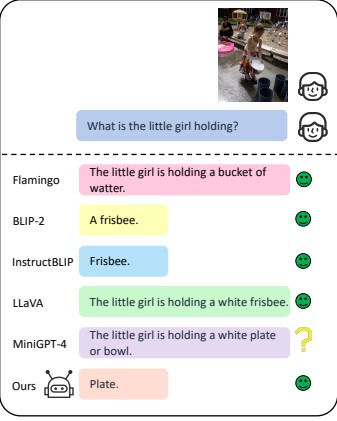
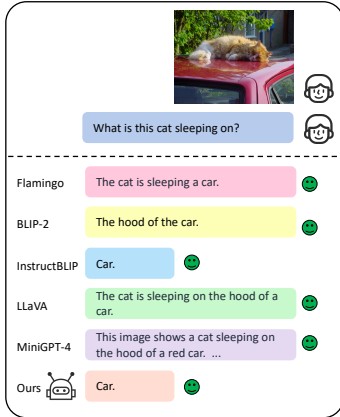

Figure 3: Qualitative comparison of different MLLMs on the object classification task. Because the task is relatively simple, MLLMs can answer those questions correctly.

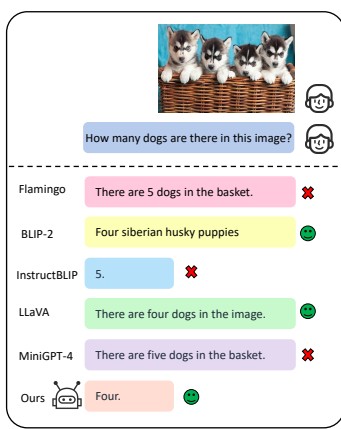
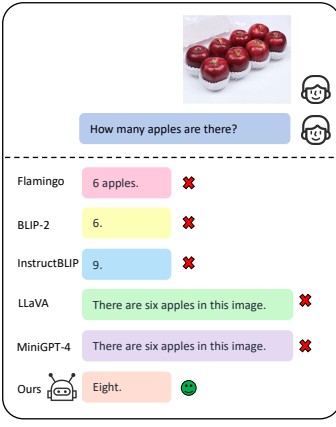
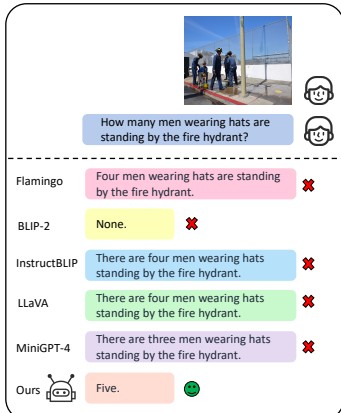

Figure 4: Qualitative comparison of different MLLMs on the counting task. Because the task requires the adapter correctly uncover visual context and shift the attention to the regions of the counting target, prompt-unaware methods may misunderstand the scene and thus make mistakes.

## 4.2 QUALITATIVE EVALUATION

Besides the systematic evaluation on four benchmarks, in this section, we further qualitatively examine the proposed method with more diverse images and prompts.

**Perception Ability.** First, we compare the visual question answering performance of five popular MLLMs, including Flamingo (Alayrac et al., 2022), BLIP-2 (Li et al., 2023b), InstructBLIP (Dai et al., 2023), LLaVA (Liu et al., 2023) and MiniGPT-4 (Zhu et al., 2023), on four perceptual tasks. Although the design of prompt may have a significant influence on output, all models are evaluated under the same unified prompt for pair comparison. As shown in Figs. 3∼6, we visualize the question-answering process of MLLMs on four perception tasks. It can be observed that the compared MLLMs all perform well in the object classification task. However, they still have room for improvement in the other three perception tasks (especially counting task). Thanks to the prompt-

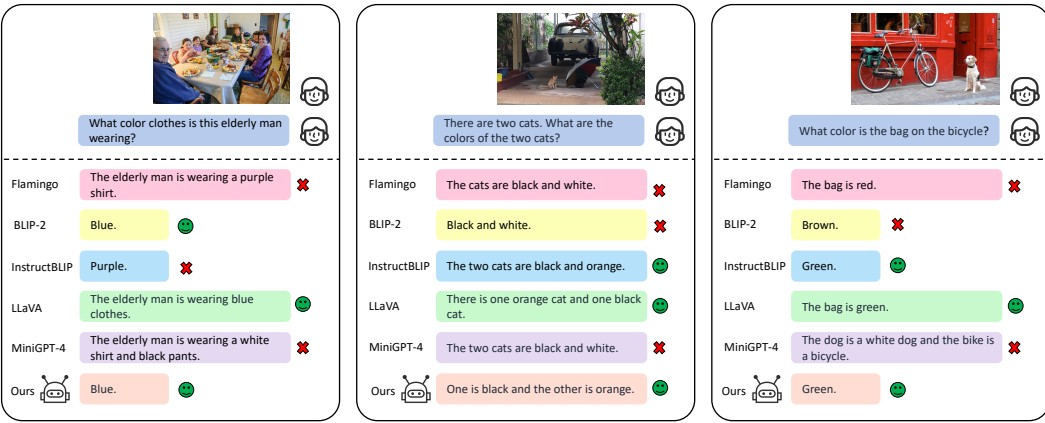

Figure 5: Qualitative comparison of different MLLMs on the color recognition task. Although the task is not that challenging, some methods may not correctly capture the target object, thus leading to wrong answers.

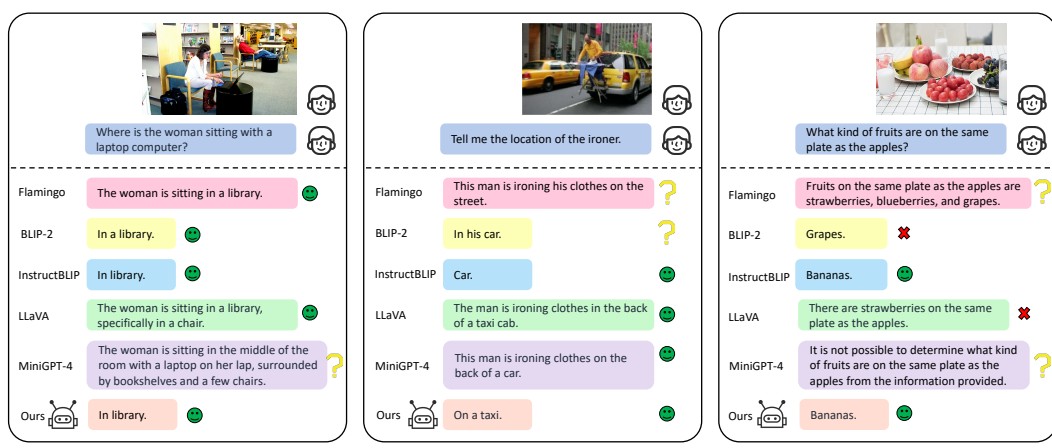

Figure 6: Qualitative comparison of different MLLMs on the position reasoning task. Existing methods can correctly answer coarse-grained position questions, such as big scenes (library), but may fail to answer fine-grained position questions, such as small objects.

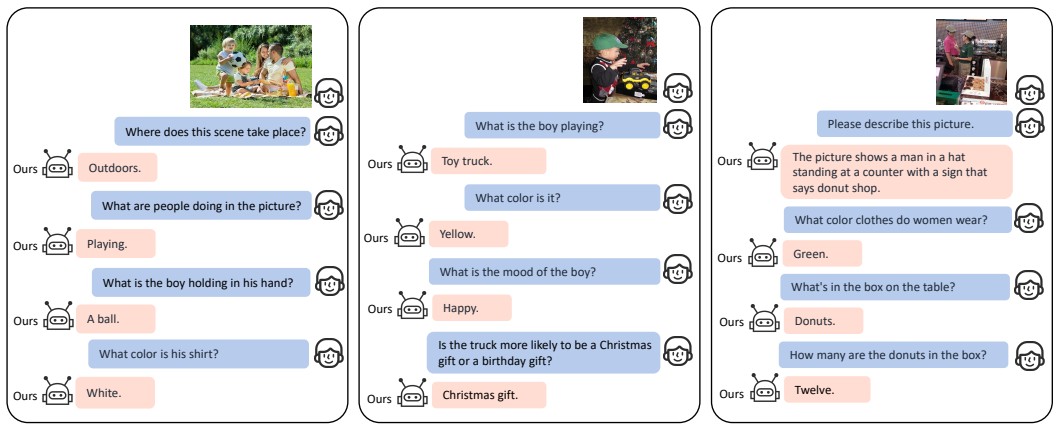

Figure 7: Visualization of multi-round dialogues.

aware adapter in our model, there is a noticeable improvement in visual perception, as it allows for a more focused attention on objects mentioned in the question. Second, through multiple rounds

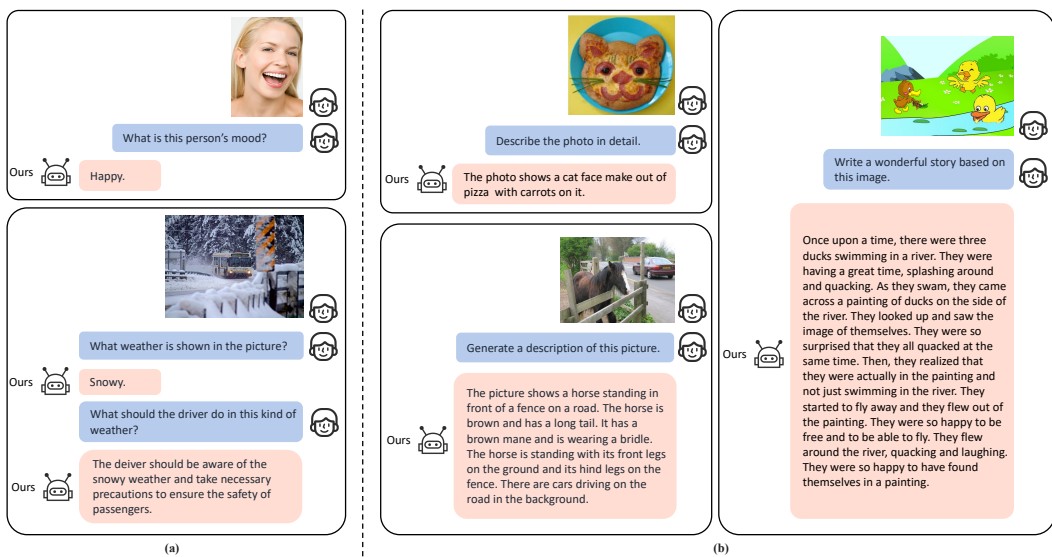

Figure 8: Visualization of the (a) cognitive and (b) generalization ability of the proposed method.

of dialog, we evaluate the ability of the proposed model to process contextual information. Fig. 7 indicates that the model is capable of capturing and utilizing information from the dialogue history.

**Cognitive Ability.** Fig. 8 (a) demonstrates the cognitive ability of our model. We find that it has acquired certain common-sense and is able to reason based on the knowledge.

**Generalization Ability.** The zero-shot generality of MLLMs depends on the diversity of tasks involved during training. To assess the generalization ability of the model, we provide more visualization results for zero-shot image-to-text generation in Fig. 8 (b) and Appendix A.4.

## 4.3 LIMITATIONS AND FUTURE WORK

Despite the MLLM equipped with a prompt-aware adapter exhibits enhanced visual perception and reasoning capabilities, it still faces certain limitations. First, MLLMs do not adhere to prompts as well as LLMs. For instance, the length of generated sentences is strongly correlated with the training data, and the specified sentence length in the prompt are often ignored. Hence, future research can explore making MLLMs instruction-sensitive by constructing high-quality and diverse multi-modal instructional datasets. Second, we find that the model has lost some common-sense knowledge. In future work, we consider replacing the fine-tuning approach by incorporating prior knowledge in the prompt to intuitively guide the model.

## 5 CONCLUSIONS

In this paper, we propose a new family of adapters for MLLMs, prompt-aware adapter, which is equipped with an ability of dynamically embedding visual inputs based on the prompt. The proposed adapter aims at extracting the most informative visual clues to the prompt and facilitating LLMs for visual understanding. Then, we implement a prompt-aware adapter with learnable query embeddings, a visual transformer and a textual transformer. In consequence, the generated prompt-aware visual tokens can largely alleviate the visual perception burden of LLMs. To evaluate the proposed prompt-aware adapter, we conduct experiments on various questions, including object classification, color recognition, counting and position reasoning. Experimental results demonstrate that the proposed method achieves outstanding performance on multiple visual perception and cognitive tasks.

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

# A  APPENDIX

## A.1  PROMPT TEMPLATES

The prompt templates employed in the end-to-end prompt fine-tuning process across various tasks are illustrated in Table 3.

Table 3: Prompt templates used for converting datasets into prompt-tuning data. In visual question answering tasks, two formats of prompts are available. Note that the "{Question}" format is mandatory for the prompt-aware adapter, and needs to be replaced with the specific question in Table 4.

| Task | Prompt Template |
|---|---|
| Image Captioning | *###Human: <ImageHere></Img> Describe this image in detail. ###Assistant:*
*###Human: <ImageHere></Img> Take a look at this image and describe what you notice. ###Assistant:*
*###Human: <ImageHere></Img> Please provide a detailed description of the picture. ###Assistant:*
*###Human: <ImageHere></Img> Could you describe the contents of this image for me? ###Assistant:*
*###Human: <ImageHere></Img> this image common in real world? ###Assistant:* |
| Visual Question Answering | *###Human:<ImageHere></Img> Describe this image in detail. ###Assistant:*
*###Human: <ImageHere></Img> Take a look at this image and describe what you notice. ###Assistant:*
*###Human: <ImageHere></Img> Please provide a detailed description of the picture. ###Assistant:*
*###Human: <ImageHere></Img> Could you describe the contents of this image for me? ###Assistant:*
*###Human: <ImageHere></Img> Is this image common in real world? ###Assistant:*
*###Human: <ImageHere></Img> "{Question}" ###Assistant:* |

## A.2  TASKS BASED ON COCO-QA

**Object Classification.** The primary objective of the object classification task is to accurately recognize objects in the given visual input and determine the category to which they belong. The metric used to evaluate this task is accuracy. For this task, we utilize $82,198$ question-answer pairs related to objects from the COCO-QA dataset (Ren et al., 2015), with $78,666$ used for fine-tuning and $3,532$ for testing. The questions typically involve the use of "what" to inquire about the object type. In addition, a succinct single-word answer is employed to precisely indicate the response.

**Counting.** The main goal of counting is to recognize and count the number of questioned objects from the given visual input. The metric employed to assess this task is accuracy. In the counting task, we use $19,568$ question-answer pairs related to the number of objects from the COCO-QA dataset (Ren et al., 2015), with $1,8761$ used for fine-tuning and $807$ for testing. When posing questions, the inquiry typically involves the use of "how many" to inquire about the object amount. Likewise, we only use a single word describing the quantity in the response.

**Color Recognition.** The color recognition task is designed to detect objects in questions and perceive color information from the input visual signal. The metric employed to assess this task is also accuracy. In this task, we utilize $8,640$ question-answer pairs related to objects from the COCO-QA dataset (Ren et al., 2015), with $8,304$ used for fine-tuning and $336$ for testing. Questions related to color are relatively straightforward, usually beginning with "What is the color of". To facilitate quantitative evaluation, responses responses are kept as brief as possible for color-related words.

**Position Reasoning.** In the position reasoning task, our primary goal is to infer the location information of the queried object based on the input visual content. Similarly, we adopt the accuracy as the evaluation metric. We adopt $7,278$ question-answer pairs related to objects from the COCO-QA dataset (Ren et al., 2015), with $6,953$ used for fine-tuning and $325$ for testing. In this task, questions are formatted to start with "where" to ask for the position of an object. Answers mostly be places, scenes, or large objects that contain smaller objects.

Examples of questions and answers from four perceptual tasks are shown in Table 4.

## A.3  MULTIPLE MODEL ARCHITECTURE IMPLEMENTATION

In the implementation of prompt-aware adapter, we try to add text cross-attention to different attention blocks. Table 5 shows the quantitative evaluation results of various model specific implementations.

Table 4: Examples of questions and answers from four visual perception tasks.

| Visual Perception Task | Question | Answer |
|---|---|---|
| Object Classification | *what are sitting down on the ground*
*what is parked on the side of the grass*
*what are two men playing with some elephants*
*what is the color of the shirt*
*what is laying on the bed next to some pillows* | *bears*
*motorcycle*
*ball*
*blue*
*cat* |
| Counting | *how many men is sitting on the street in front of a building*
*how many red velvet cup cakes with no frosting on a flowered plate*
*how many pairs of shoes on a mat with a cat is sitting in the middle*
*how many dessert treats in the white cardboard box*
*how many trays of itallian food are in large pans* | *two*
*three*
*eight*
*six*
*four* |
| Color Recognition | *what is the color of the airplane*
*what is the color of the motorcycle*
*what is the color of the brush*
*what is the color of the bird*
*what is the color of the flowers* | *black*
*orange*
*green*
*white*
*red* |
| Position Reasoning | *where is the cat lounging*
*where do the mother and son make sundaes*
*where is the cheese pizza*
*where is the person sitting*
*where do the large and over-sized stuffed teddy bear sitting* | *chair*
*kitchen*
*box*
*bed*
*chair* |

Table 5: Quantitative evaluation results of various model specific implementations. The "No." is the number of the experimental setup. The "BlockNum" denotes the number of attention block in which the text cross-attention layer is inserted. The "OC", "CON", "CR", and "PR" represent the four tasks of object classification, counting, color recognition and position reasoning, respectively. The "Acc (%)" denotes accuracy and "Total Acc (%)" refers to the overall accuracy on four tasks.

| No. | BlockNum | OC
Acc (%) | CON
Acc (%) | CR
Acc (%) | PR
Acc (%) | Total
Acc (%) |
|---|---|---|---|---|---|---|
| 1 | 0 | 78.09 | **73.22** | 74.23 | 65.54 | 76.33 |
| 2 | 0+1 | 65.46 | 70.84 | 51.55 | 52.31 | 62.73 |
| 3 | 0+1+2 | 77.67 | 65.48 | 73.49 | 63.08 | 75.23 |
| 4 | 0+1+2+3 | 78.20 | 69.95 | 69.89 | 62.47 | 75.29 |
| 5 | 0+1+2+3+4 | 77.72 | 69.35 | 79.06 | 62.47 | 76.38 |
| 6 | 0+1+2+3+4+5 | 78.52 | 71.43 | 74.85 | 67.39 | 76.73 |
| 7 | 1 | 29.31 | 36.31 | 8.68 | 10.47 | 25.23 |
| 8 | 1+3+5+7+11 | 33.21 | 29.47 | 9.79 | 22.77 | 28.50 |
| 9 | 0+2+4+6+8+10 | 77.98 | 69.05 | 72.37 | 64.31 | 75.59 |
| 10 | 11 | 77.02 | 68.46 | 60.97 | 62.77 | 72.93 |
| 11 | 10+11 | 75.15 | 71.73 | 72.37 | 64.62 | 73.79 |
| 12 | 9+10+11 | 74.89 | 69.95 | 72.87 | 66.47 | 73.69 |
| 13 | 8+9+10+11 | 75.09 | 69.95 | 73.49 | 65.85 | 73.89 |
| 14 | **7+8+9+10+11** | **78.65** | 72.03 | **78.31** | **67.69** | **77.26** |
| 15 | 6+7+8+9+10+11 | 77.72 | 68.76 | 75.97 | 64.92 | 76.1 |
| 16 | 0+1+2+3+4+5+6+7+8+9+10+11 | 76.70 | 69.05 | 74.73 | 64.62 | 75.09 |

## A.4 MORE CASE STUDIES

In this section, we give more visualization results in Figs. 9∼10.

## A.5 EXPERIMENTAL FINDINGS

**It is crucial to strike a balance between accuracy and language generation capability.** We find that, on the one hand, insufficient training on downstream tasks leads to issues such as hallucination and persistent errors. On the other hand, excessive fine-tuning in downstream tasks hinders the

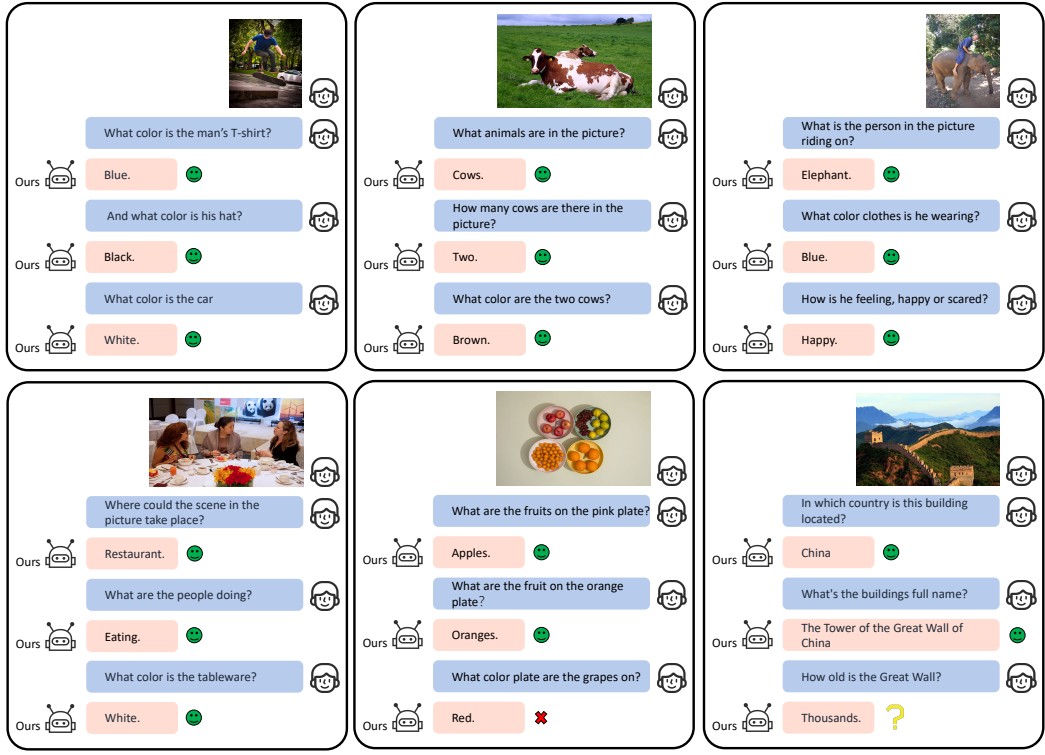

Figure 9: Selected examples of zero-shot image-to-text generation using a MLLM base on prompt-aware adapter, where it shows favorable visual perception ability.

ability of the model to comply with user prompts for generating long sentences. Besides, over-training of MLLMs can lead to catastrophic forgetting of the original knowledge of LLMs.

**Tasks and prompts are crucial for zero-shot abilities.** We observe that the responses of MLLM tend to lean towards the prompts and tasks encountered during the training process. That means diverse prompts have a significant impact on the final performance of MLLMs.

**MLLMs do not follow prompts as well as LLMs.** We discover that different MLLMs have their own preferences for the length of generated sentences, rather than strictly following prompts. For example, regardless of the input prompt, our model, similar to InstructBLIP (Dai et al., 2023), tends to generate short responses. In contrast, other models, such as LLaVA (Liu et al., 2023) and MiniGPT-4 (Zhu et al., 2023), tend to generate longer sentences without considering the prompt, such as "give a short answer" or "answer in one word."

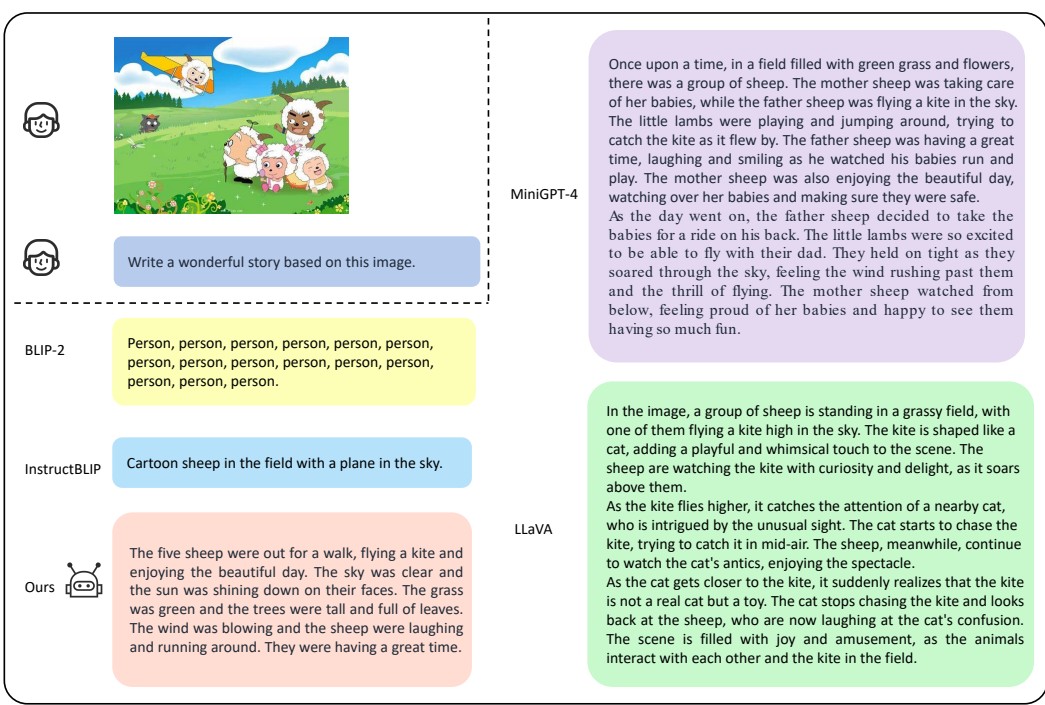

Figure 10: Comparison of multiple MLLMs for generating descriptions of images. BLIP-2 is unable to generate requested content, and InstuctBLIP tends to generate short descriptions. MiniGPT-4 and LLaVA tend to generate large paragraphs of text. Our method generates text of moderate length.

