# OpenReview forum: "Prompt-aware Adapter: Towards Learning Effective Visual Tokens for GPT4-Style Multimodal Models"
_ICLR.cc/2024/Conference — ICLR 2024 Conference Withdrawn Submission_

### Official Review · Reviewer_3AJJ · 2023-10-29

**Soundness:** 3 good
**Presentation:** 3 good
**Contribution:** 1 poor
**Rating:** 3
**Confidence:** 4

**Summary:**

This work focuses on vision language models and proposes prompt-aware adapters. It utilizes a similar framework with InstructBLIP, which utilizes an instruction-guided adapter for text-aware visual tokens. Experiments show part of the effectiveness.

**Strengths:**

1. This work focuses on the instruction-guided method for the vision language model, which is promising.
2. The proposed method is simple and easy to follow.
3. The presentation is clear.

**Weaknesses:**

1. The proposed method is very similar to InstructionBLIP from the framework to the training strategy. This work also utilizes the pre-trained QFormer in BLIP. It is essential to reveal the difference between this work and InstructBLIP. This strongly harms the technical contribution of this work.
2. Comparisons with previous work are missing. It is necessary to compare it with other vision language models (especially InstructBLIP) on public datasets, like VQAV2, GQA, ScienceQA, COCO Caption, et.al.

**Questions:**

Please refer to the weakness section.

---

### Official Review · Reviewer_1W2T · 2023-10-30

**Soundness:** 2 fair
**Presentation:** 3 good
**Contribution:** 2 fair
**Rating:** 5
**Confidence:** 3

**Summary:**

The paper aims to improve the conversion of visual inputs into relevant tokens for GPT4-style multimodal large language models (MLLMs) by addressing the limitations of current adapters.

**Proposed Solution:**
A "prompt-aware adapter" is introduced, designed to dynamically embed visual inputs based on textual prompts, thus optimizing visual information extraction.

**How it Works:**
The adapter utilizes textual prompts as a guide during conversion, employs learnable query embeddings, a visual transformer, and a textual transformer to interact with visual features based on prompts.

**Evaluation & Results:**
Tests on various visual tasks showed improved performance with the prompt-aware adapter, increasing accuracy across tasks by 2.89%, 5.80%, 1.43%, and 2.98%.

The new prompt-aware adapter enhances the visual reasoning of MLLMs by focusing on relevant visual cues based on textual prompts. The code for the research will be made public.

**Strengths:**

1. **Innovation & Addressing a Gap:** The paper introduces a novel "prompt-aware adapter" that optimally converts visual inputs to relevant tokens for GPT4-style models, highlighting and addressing current model limitations.

2. **Practical Implementation & Validation:** The research provides a comprehensive description of the adapter's practicalities and validates its effectiveness through experiments, showcasing improved accuracy across multiple visual tasks.

**Weaknesses:**

**Weaknesses of the Paper:**

1. **Simplicity of Method:** The method appears straightforward, with the primary approach being the direct addition of text embedding in the adapter.

2. **Limited Trainable Parameters:** The paper does not address the issue of having too few trainable parameters, which raises questions about its capacity to handle complex tasks.

3. **Questionable Integration:** The decision to process text and visual attention operations layer-by-layer in the Prompt-Aware Adapter, instead of directly concatenating them, is not clearly justified.

4. **Overemphasis on Common Knowledge:** A substantial portion of the content reiterates common knowledge, such as the formulas for attention, which may be redundant for seasoned readers.

5. **Limited Experimental Scope:** Experiments were primarily conducted on the smaller scale MiniGPT-4 model. The paper lacks quantitative comparisons with state-of-the-art (SOTA) image-text understanding models like Lava and Flamingo.

6. **Unclear Multimodal Context Handling:** It's unclear how the adapter would function in multi-turn dialogues when only textual input is present.

7. **Potential Bias in Illustrations:** Figure 6 presents selective examples, which might be cherry-picking, thus not offering a clear advantage over comparative methods.

**Questions:**

Pls see the weakness.

---

### Official Review · Reviewer_p8Xv · 2023-10-31

**Soundness:** 4 excellent
**Presentation:** 3 good
**Contribution:** 2 fair
**Rating:** 5
**Confidence:** 4

**Summary:**

This paper proposes a prompt-aware adapter for multimodal large language models (MLLM). Instead of simply mapping visual tokens into understandable tokens of LLMs via a linear layer, the prompt-aware adapter adopts a DETR decoder-like structure to aggregate information from prompts, easing the visual parsing of LLMs. The method is simple but effective across various tasks.

**Strengths:**

The motivation is clear, each time the prompt is switched (different tasks), the visual input to LLMs stays stationary, a natural way to improve it is to make it prompt-aware, i.e., conditioning the visual input to LLMs on prompts. The experiment results show remarkable improvement even over finetuning the Q-Former.

**Weaknesses:**

The chosen tasks are relatively low-level, they hardly need the reasoning capability of LLMs, could authors provide more experiments on high-level tasks to prove the prompt-aware adapter's generalization ability?

**Questions:**

Is there any better prompt-aware adapter structures than this query-based decoder, or should the main focus of this paper put on designing the prompt-aware structure since this dynamic idea is prevalent across computer vision and exist for years ?

---

### Official Review · Reviewer_PKd5 · 2023-10-31

**Soundness:** 2 fair
**Presentation:** 2 fair
**Contribution:** 2 fair
**Rating:** 3
**Confidence:** 4

**Summary:**

This paper proposes a prompt-aware adapter for multimodla LLM, which fuses visual and textual information as the input of LLM. Preivous works take visual information as the input to LLM, ignoring the relation between visual and textual information. The proposed method leverage a smaller language model as prompt adapter to extract prompt relevant information so that the LLM can better focus on the human instructions. The proposed method is evaluated on COCO-QA dataset with better performance than another MLLM, MiniGPT4.

**Strengths:**

- The motivation is reasonable and the method is clearly explained.
- The proposed method is evaluated on several tasks with promising results.

**Weaknesses:**

- The prompt adapter is nearly the same as Language-Guided Tokenizer in VisionLLM [1]. They all take the visual feature and text prompt as  input and output fused visual tokens. Both Language-Guided Tokenizer in VisionLLM and prompt adapter in this work utilize Bert the basic model.
- The evaluation is very week. The proposed method has not evaluated on either traditional captioning, VQA tasks and recent multimodal benchmarks [2,3,4]. Beisdes, the quantitative comparison baseline is just MiniGPT4. More comparison is needed.
- More ablation experiments are needed here to prove that prompt adapter is really helpful.

[1] Wang, Wenhai, et al. "Visionllm: Large language model is also an open-ended decoder for vision-centric tasks." arXiv preprint arXiv:2305.11175 (2023).
[2] Fu, Chaoyou, et al. "MME: A Comprehensive Evaluation Benchmark for Multimodal Large Language Models." arXiv preprint arXiv:2306.13394 (2023).
[3] Liu, Yuan, et al. "MMBench: Is Your Multi-modal Model an All-around Player?." arXiv preprint arXiv:2307.06281 (2023).
[4] Li, Bohao, et al. "Seed-bench: Benchmarking multimodal llms with generative comprehension." arXiv preprint arXiv:2307.16125 (2023).

**Questions:**

n/a.